# Nanoscale tomography reveals the deactivation of automotive copper-exchanged zeolite catalysts

Joel E. Schmidt [1], Ramon Oord [1], Wei Guo [2], Jonathan D. Poplawsky [2] & Bert M. Weckhuysen [1]

Copper-exchanged zeolite chabazite (Cu-SSZ-13) was recently commercialized for the selective catalytic reduction of $NO_X$ with ammonia in vehicle emissions as it exhibits superior reaction performance and stability compared to all other catalysts, notably Cu-ZSM-5. Herein, the 3D distributions of Cu as well as framework elements (Al, O, Si) in both fresh and aged Cu-SSZ-13 and Cu-ZSM-5 are determined with nanometer resolution using atom probe tomography (APT), and correlated with catalytic activity and other characterizations. Both fresh catalysts contain a heterogeneous Cu distribution, which is only identified due to the single atom sensitivity of APT. After the industry standard 135,000 mile simulation, Cu-SSZ-13 shows Cu and Al clustering, whereas Cu-ZSM-5 is characterized by severe Cu and Al aggregation into a copper aluminate phase ($CuAl_2O_4$ spinel). The application of APT as a sensitive and local characterization method provides identification of nanometer scale heterogeneities that lead to catalytic activity and material deactivation.

---

[1] Debye Institute for Nanomaterials Science, Utrecht University, Universiteitsweg 99, 3584 CG Utrecht, The Netherlands. [2] Center for Nanophase Materials Sciences, Oak Ridge National Laboratory, Oak Ridge, TN 37831, USA. Correspondence and requests for materials should be addressed to J.D.P. (email: poplawskyjd@ornl.gov) or to B.M.W. (email: B.M.Weckhuysen@uu.nl)

In modern diesel vehicles, meeting the mandated goals of decreased fuel consumption, particulate and $NO_X$ emissions requires end-of-tailpipe technologies due to their interconnected nature[1]. Although several strategies exist for mobile $NO_X$ reduction, by far the most effective is ammonia selective catalytic reduction ($NH_3$-SCR), with urea serving as a source of the ammonia reductant, and the reaction occurring over a copper (Cu)-exchanged zeolite catalyst[2–4]. Mobile SCR catalysts face formidable requirements including stable operation across low (engine start) and high (hydrocarbon contaminant burn off) temperature regimes, resistance to poisoning and long lifetimes[5]. Cu-exchanged zeolite ZSM-5 (further denoted as Cu-ZSM-5) was first described for SCR in 1986[6], but is unable to meet lifetime requirements due to deactivation under tailpipe conditions[7–9]. A true success story of recent zeolite catalysis was the discovery of Cu-exchanged zeolite SSZ-13 (further denoted as Cu-SSZ-13) for mobile $NO_X$ SCR with ammonia (further denoted as $deNO_X$) in the mid-2000s, followed by its rapid commercialization in 2010, due to its high activity and stability[5,10–12].

Zeolite SSZ-13 is a small-pore zeolite with the chabazite (CHA) framework, containing cages that are limited by 8-membered ring (8 MR) windows (~3.7 Å), and the silicoaluminophosphate composition of this structure, i.e., SAPO-34, is currently applied commercially for the methanol-to-hydrocarbons (MTH) reaction[13]. The superior performance of Cu-SSZ-13 as a mobile $deNO_X$ catalyst is ascribed to its small pores preventing the admission of contaminants and Al migration, the double six-membered rings (d6r's) serving as the preferential Cu exchange site, and the intrinsic stability of the framework structure[14–29]. Deactivation of Cu-exchanged zeolite catalysts has been studied by numerous characterization techniques, and is thought to occur through a combination of loss of Al from the zeolite framework and Cu migration to form Cu oxides and Cu aluminate species, similar to a $CuAl_2O_4$ spinel phase[5,14,15,21,24,30–34]. For Cu-ZSM-5, it is known that the deactivated material retains crystallinity, despite loss of framework Al which is able to move through the 10 MR pores, as well as migration and sintering of Cu, whereas Cu-SSZ-13 is not deactivated under identical conditions[7,8,23,27,34–36]. However, there still remain significant gaps in our understanding of the exact deactivation mechanisms in these materials due to the small size of deactivating species, which makes them invisible to many techniques, e.g., X-ray diffraction, and the presence of paramagnetic Cu which can interfere with locally sensitive techniques, e.g., NMR. Therefore, it is necessary to reconstruct the distribution of single atoms in 3D to gain a complete picture of deactivation.

Here, we show the material changes upon aging by creating atom-by-atom reconstructions of both fresh and aged, industrially relevant zeolite catalysts using atom probe tomography (APT). Zeolites Cu-ZSM-5 and Cu-SSZ-13 are aged using the industry standard protocol to simulate 135,000 miles of vehicle use, and characterized using the conventional techniques of powder X-ray diffraction (XRD), ammonia temperature-programmed desorption ($NH_3$-TPD), UV–Vis-NIR diffuse reflectance spectroscopy (UV–Vis-NIR-DRS) and $deNO_X$ reaction testing[28]. The distributions of Al and Cu are mapped with nanometer resolution in 3D using APT. Heterogeneities in both the Cu and Al distributions are found in all four catalysts (i.e., fresh and aged Cu-ZSM-5 and Cu-SSZ-13), and the significant changes between the fresh and aged materials can account for the observed differences in $deNO_X$ reactivity, as is schematically shown in Fig. 1.

## Results

**Catalyst characterization and reactivity.** The four Cu-exchanged zeolite materials used in this work are typical for commercially relevant $deNO_X$ catalysts[12,28], with details on their preparation and characterization given in the Methods and Supplementary Methods. The results of the $deNO_X$ reaction testing are given in

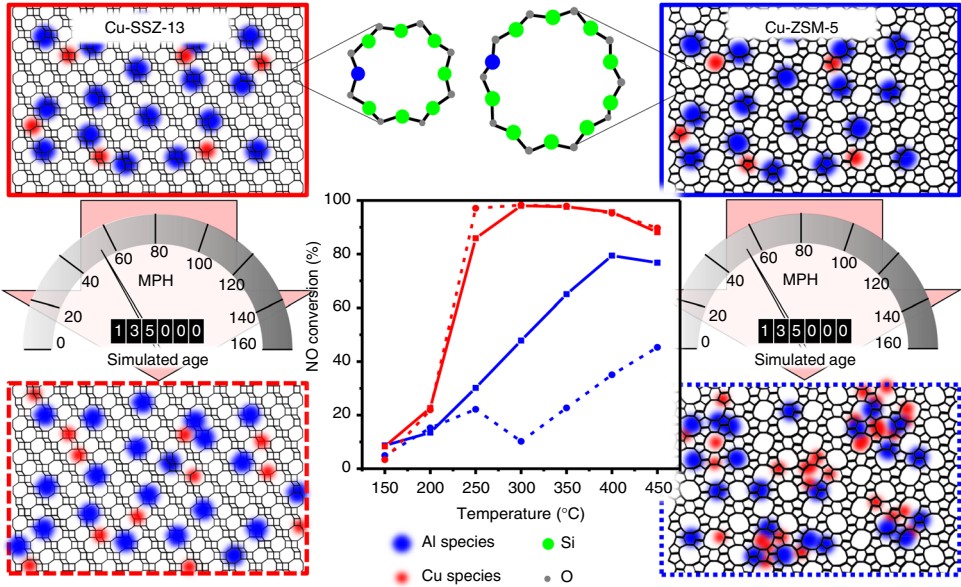

**Fig. 1** Reaction testing and overview of experimental findings. The center graph shows the NO conversion for the ammonia selective catalytic reduction ($NH_3$-SCR) reaction testing results for the four Cu-exchanged zeolite materials under study, the traces correspond to the borders of the four schematics. The feed consists of 1000 ppm NO, 1000 ppm $NH_3$, 5% $O_2$, balanced by He. The data have been measured in steady-state mode (GHSV = 100,000 h$^{-1}$). The panels at the corners give a 2D schematic overview of the results of this study with regards to the distribution of Al and Cu in Cu-SSZ-13 and Cu-ZSM-5, which are distributed slightly heterogeneously in fresh Cu-SSZ-13 and Cu-ZSM-5. After the simulated 135,000 mile aging procedure the Cu in Cu-SSZ-13 is found to migrate somewhat, as the schematic shows, though the material still retains good catalytic activity. In Cu-ZSM-5, the simulated aging causes significant migration of Cu and Al into stoichiometric copper aluminate species. One of the significant reasons that Cu-ZSM-5 is more damaged is that both Cu and Al species are able to migrate through its larger rings, whereas the small rings in Cu-SSZ-13 prevent Al migration, limiting damage to the material

Fig. 1, and show that zeolite Cu-SSZ-13 retains its activity even after aging, whereas zeolite Cu-ZSM-5 shows a significant reduction in deNO$_X$ activity after aging, in line with the results of other studies[5,7–12]. The XRD patterns of the fresh and aged Cu-ZSM-5 and Cu-SSZ-13 materials are shown in Supplementary Figs. 1 and 2, and in all four samples no significant framework destruction or peaks associated with Cu oxides or CuAl$_2$O$_4$ spinel could be observed, meaning that the zeolite frameworks remain intact after aging and that Cu is well-dispersed and is not agglomerated, at least at length scales that the XRD method is capable of detecting. The inability to detect material changes with XRD reinforces the need for advanced characterization techniques to truly elucidate deactivation phenomena. UV–Vis–NIR-DRS measurements show a significant difference between the two Cu-exchanged zeolite materials, with further changes after aging that indicate possible CuO and CuAl$_2$O$_4$ spinel species, shown in Supplementary Figs. 3 and 4, along with a detailed explanation of the interpretation of the spectra in the Supplementary Methods. NH$_3$-TPD (Supplementary Methods; Supplementary Fig. 5) shows the expected changes with aging, with a decrease in strong acid sites attributed to framework Al removal, and Cu ions present in all samples[18]. Although these bulk techniques indicate changes to the material after aging, they offer a global view of a nanoscopic phenomena, motivating our study.

An unanswered question, critical to understanding the deactivation of these catalysts, is the fate of Al and Cu after removal from the framework. Chemically or spatially resolved insights regarding the deactivation of Cu-exchanged zeolites have been gained using X-ray absorption fine structure (XAFS), transmission electron microscopy (TEM), scanning TEM

(STEM), STEM electron dispersion spectroscopy (STEM-EDS), and X-ray photoelectron spectroscopy (XPS), but these techniques suffer from significant drawbacks including the inability to detect isolated ions, as well as offering only 2D information or bulk averages[14,24,28,30–34,37,38]. To gain a more complete picture of the mechanism of deactivation in these materials, and establish the distribution of both Cu and framework elements (e.g., Al and Si), we studied all four samples using APT, and differences in the elemental distributions are correlated with both spectroscopy and reactivity. A recent study on Fe-SSZ-13 used APT to characterize the material before and after aging, with the conclusion that the Fe was randomly distributed and not clustered in either sample, but the APT results were in disagreement with the findings of TEM/EDS, showing the need for a more comprehensive study of the commercialized Cu-SSZ-13 catalyst[39]. A potential challenge in studying Fe-SSZ-13 is that there is overlap between the primary Fe ($^{56}$Fe$^{2+}$) and Si ($^{28}$Si$^+$) peaks in the mass spectrum, potentially complicating the quantification and subsequent analysis, but no similar overlap is encountered in the present work with Cu. Full details of APT sample preparation, experiment and data analysis are given in the Methods and Supplementary Methods, and SEM images of both catalysts before and after being fashioned into needles for APT analysis are shown in Supplementary Fig. 6. When the needles were reconstructed, compositional heterogeneities were found that spanned a range of length scales, which can account for the differences in deNO$_X$ reaction behavior. The following sections will be structured by first discussing Cu-ZSM-5 and then Cu-SSZ-13, followed by a comparison of the two materials, and Supplementary Table 1 lists all samples along with the

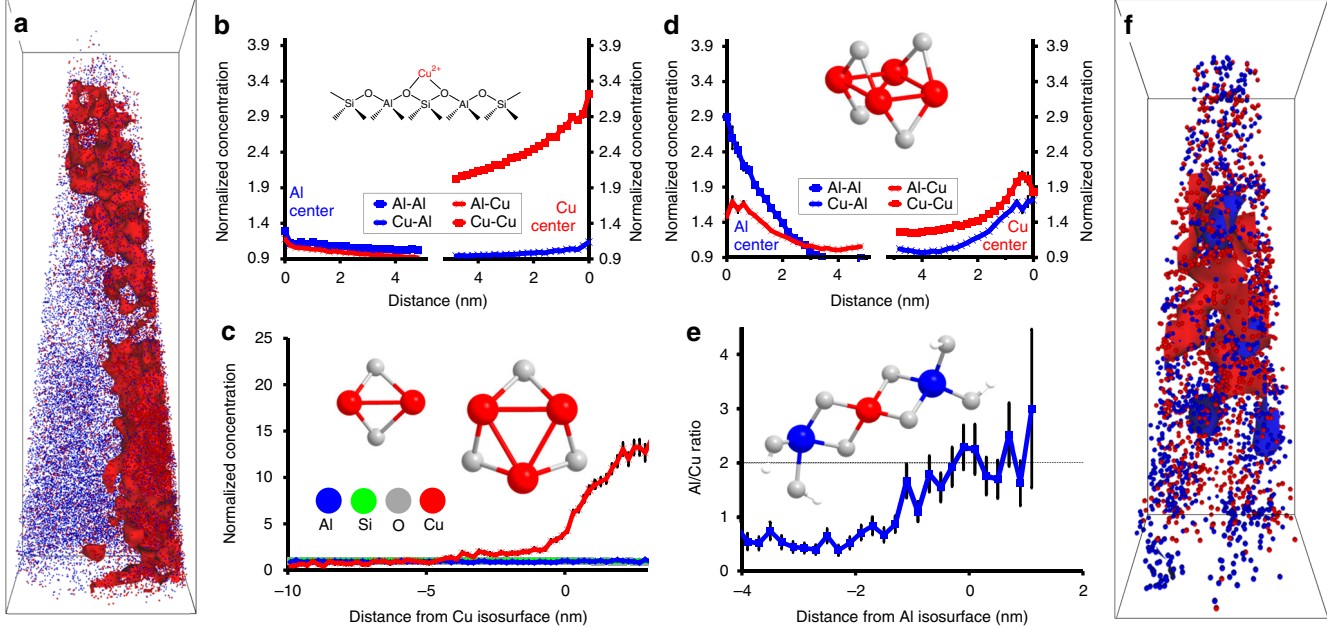

**Fig. 2** Overview of atom probe tomography results for fresh and aged Cu-ZSM-5. **a** Reconstructed needle of fresh Cu-ZSM-5 (**1**, full results are in Supplementary Fig. 22) with Cu (red) and Al (blue) ions shown and 1.8% Cu isoconcentration surfaces shown, this is also provided as Supplementary Movie 1. Bonding box dimensions are 40 × 43 × 140 nm$^3$. **b** Radial distribution functions (RDFs) in fresh Cu-ZSM-5 for Al and Cu centers. Strong Cu–Cu affinity is indicated, as well as much weaker, but still significant affinities between all other species. An Al–Cu affinity is expected due to Cu exchanging onto paired Al sites, as indicated in the schematic. **c** Normalized compositional histogram across the 1.8% Cu isoconcentration surfaces in fresh Cu-ZSM-5, with the structures of potential CuO species shown. **d** RDFs in aged Cu-ZSM-5 (**3**, full results are in Supplementary Fig. 26) for Al and Cu centers, with the structure of a potential CuO species shown. Strong affinities are indicated between all species, pointing to the formation of copper aluminate domains after aging due to Cu–Al aggregation. **e** Al/Cu ratio across the 8% Al isoconcentration surfaces in aged Cu-ZSM-5, the stoichiometry of Al/Cu = 2 is consistent with CuAl$_2$O$_4$ spinel, and the O/Cu and O/Al values are also stoichiometric, see Supplementary Fig. 27. **f** Reconstructed needle of aged Cu-ZSM-5 (**3**) with 5% Cu and 8% Al isoconcentration surfaces shown, this is also included as Supplementary Movie 3. Bonding box dimensions are 15 × 15 × 43 nm$^3$. All error bars were calculated from counting statistics using the method described in ref. [55]

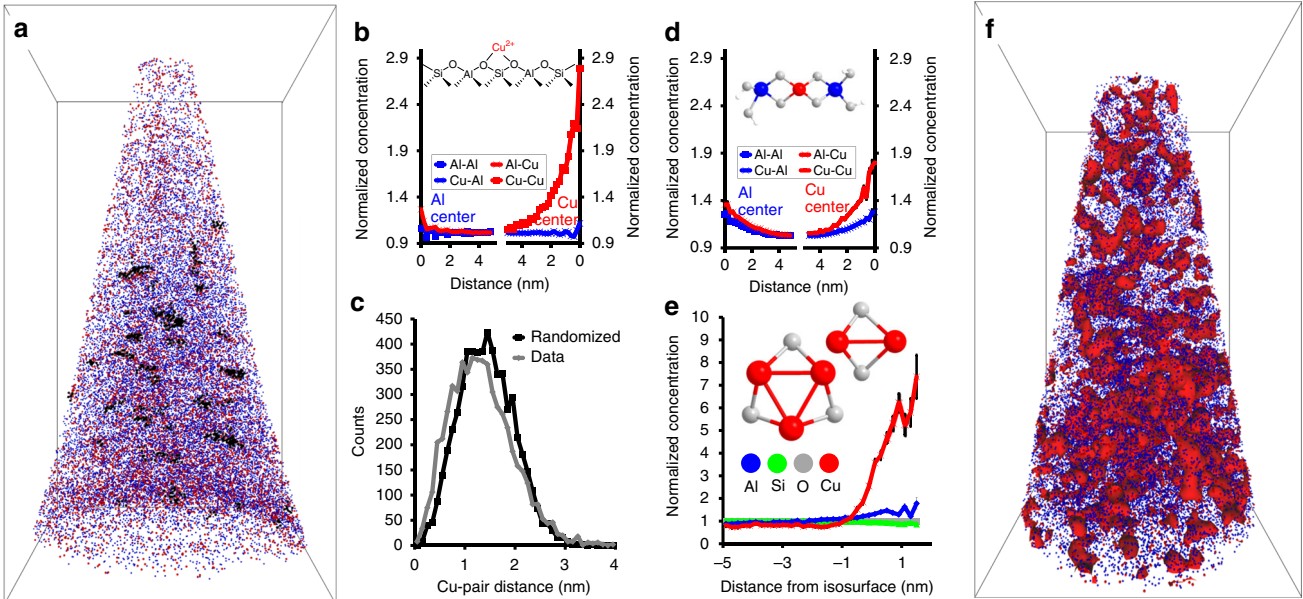

**Fig. 3** Overview of atom probe tomography results for fresh and aged Cu-SSZ-13. **a** Reconstructed needle of fresh Cu-SSZ-13 (**4**, full results are in Supplementary Fig. 28) with Cu (red) and Al (blue) ions shown with Cu clusters overlaid in black, this is also included as Supplementary Movies 4 and 5. Bonding box dimensions are $63 \times 67 \times 115$ nm$^3$. **b** Radial distribution functions (RDFs) in fresh Cu-SSZ-13 for Al and Cu centers. Al–Cu affinity is indicated, which would be expected due to Cu exchanging onto paired Al sites, as indicated in the schematic. **c** Nearest neighbor distribution for fresh Cu-SSZ-13 for Cu showing a significant deviation from a random distribution and indicating the presence of Cu clusters. **d** RDFs in aged Cu-SSZ-13 (**6**, full results are in Supplementary Fig. 32) for Al and Cu centers. Strong affinities are indicated between all species, pointing to the migration and aggregation of Cu with aging, and a Cu aluminate species is shown, though it was not quantitatively identified. **e** Normalized compositional histogram across 1.4% Cu isoconcentration surfaces in aged Cu-SSZ-13, with potential CuO species shown. **f** Reconstructed needle of aged Cu-SSZ-13 (**6**) with 1.4% Cu isoconcentration surfaces shown, this is also included as Supplementary Movies 8 and 9. Bonding box dimensions are $49 \times 52 \times 90$ nm$^3$. All error bars were calculated from counting statistics using the method described in ref. [55]

compositions determined using APT, and a complete discussion of the APT analysis for each material is in the Supplementary Methods and Supplementary Figs. 7–36. In addition, Supplementary Movies 1–10 show each needle to allow for easy 3D visualization of the ion distributions and features of interest.

**Atom probe tomography of fresh and aged zeolite Cu-ZSM-5.**
Two needles of fresh Cu-ZSM-5 and one of aged Cu-ZSM-5 were successfully reconstructed as this material was especially prone to failure, with images of the fresh and aged materials shown in Fig. 2 and Supplementary Fig. 19; in all subsequent discussions of the fresh and aged materials, samples **1** to **3**, complete APT results are in Fig. 2 and Supplementary Figs. 22–27 as well as Supplementary Movies 1–3. In fresh Cu-ZSM-5, **1**, the Cu segregation was easily isolated using a 1.8 atomic %Cu isoconcentration surface (bulk Cu content is 0.4 atomic %, all subsequent compositions are in atomic percent and will be referred to by % for brevity). The compositional histogram across this isoconcentration surface shows a rapid increase in the Cu content to nearly 15 times its bulk normalized concentration. Interestingly, although the Cu content rises markedly across this region, the Al content only changes slightly, indicating that the heterogeneous Cu distribution is possibly caused by preferential material accessibility during the exchange. This Cu segregation is reflected in the Cu nearest neighbor distribution (NND) as well as the Cu radial distribution function (RDF). The Al NND for **1** also shows a small deviation from the normal distribution, which was studied using cluster analysis, with the identification of significant Al clusters shown in Supplementary Fig. 22. In our previous work using APT to study large, carefully prepared crystals of ZSM-5, the Al distribution was random in the fresh material, so finding

clustered Al in the commercial material highlights the imperfections found in industrial catalysts compared to model systems[40]. This assertion is supported by the recent application of advanced microscale and nanoscale characterization techniques, which have revealed that small, industrial zeolite crystals can be quite heterogeneous, and that the results of bulk studies give an ensemble average from a diverse population of crystals[41–45]. The second needle of fresh Cu-ZSM-5, **2**, also had Cu segregation, which is reflected in the NND and RDF. This segregation was isolated using both a 2.7% Cu isoconcentration surface and Cu cluster analysis as the regions could be described by either method. This needle had a homogeneous Al distribution, and the differences between the two needles from the same material reinforce the imperfect nature of industrial zeolite crystals, and it is possible that the heterogeneity of the material contributes to its lower activity and stability.

After aging, significant changes occurred in both the Cu and Al distributions, which are immediately apparent from ion distribution maps in Fig. 2f and Supplementary Fig. 19, with complete characterizations of sample **3** in Supplementary Figs. 26 and 27 and Supplementary Movie 3. The NND for Al shows a significant deviation from a random distribution in this material, and the Al RDF shows that there is an Al–Al affinity, as well as an Al–Cu affinity, which are caused by a Cu–Al aggregation in the sample. An 8% Al isoconcentration surface was analyzed (bulk Al content is 3.7%), and its relevance to understanding deactivation is evident when the Al/Cu ratio is considered. Across the 8% Al isoconcentration surface the Al/Cu ratio increases to ~2, which matches the stoichiometry of Cu aluminate, a CuAl$_2$O$_4$ spinel species, which has long been regarded as one of the species that forms upon aging of these materials. Previously, it was only identified by bulk analysis, and here is shown as spatially isolated

nanoscale features in 3D[14,24,31–34]. In addition, the O/Cu and O/Al values, shown in Supplementary Fig. 27, agree with their stoichiometric values inside these isoconcentration surfaces, further supporting the identification of $CuAl_2O_4$ spinel within the APT data. Upon further examination of the UV–Vis–NIR-DRS spectra (Supplementary Figs. 3 and 4), it is likely that changes upon aging are due to the presence of $CuAl_2O_4$ spinel and CuO species, though these phases are difficult to verify by spectroscopy alone due to many overlapping peaks and the influence of domain size on the peak shift, underscoring the need for advanced characterization methods to understand the formation and presence of deactivating phases. The proportion of the material contained in these Cu aluminate regions was assessed from the APT data and found to include ~20% of all Cu and ~35% of all Al (further discussed in the Supplementary Fig. 26 caption), providing a rough estimate of the loss of active sites. Therefore, we have been able to demonstrate a $CuAl_2O_4$ stoichiometry in a nanoscale region and determine the proportion of material that has transformed into this phase after aging. Data were successfully collected from a second needle of aged Cu-ZSM-5, and the results are consistent with those of needle 3, which can be found in the Supplementary Methods and Supplementary Figs. 35 and 36, though these data were not included in the main manuscript due to significant Ga implantation.

**Atom probe tomography of fresh and aged zeolite Cu-SSZ-13.** Two needles of fresh zeolite Cu-SSZ-13 were measured, with all discussed results for samples **4** and **5** shown in Fig. 3 and Supplementary Figs. 28–31 and ion maps within Supplementary Fig. 20 as well as Supplementary Movies 4–7, with only minor heterogeneities apparent from visual inspection. The Al distribution was random as no deviation was observed from the NND; this contrasts with the Cu-ZSM-5 and may be due to the different preparation methods of the zeolites. Finding a random Al distribution suggests that a Cu exchange is unlikely to cause Al segregation, so the observation of a heterogeneous Al distribution in fresh Cu-ZSM-5 is most likely not caused by ion exchange. The RDF for Al shows an Al–Cu affinity, though this would be expected as Al serves as the exchange site for Cu, shown schematically in Fig. 3b, highlighting the sensitivity of APT. The Cu distribution is not homogeneous, which is seen in the Cu NNDs for both needles, and the Cu RDFs indicate that there is a Cu–Cu affinity. Cluster analysis for Cu in these fresh materials gave statistically significant Cu clustering in both needles, and it was found that looking at the fourth nearest neighbor (O = 4) led to the best separation of Cu clusters (further discussed in the Supplementary Methods)[46,47]. The cluster analysis demonstrates that even in the fresh Cu-SSZ-13, which has only isolated Cu ions based the UV–Vis–NIR-DRS spectrum, a non-random Cu dispersion exists. It is known in small-pore zeolites that conventional aqueous ion exchange of Cu can be unpredictable and more challenging than larger pore materials, and this non-random Cu dispersion may be the result of this known difficulty[48,49]. This finding shows the Cu dispersion can be heterogeneous, though still as spectroscopically isolated Cu, giving a highly active catalyst. In addition, it highlights the power of APT to probe these materials as no other techniques can resolve isolated Cu ions in 3D[28,30,33,34,37,38].

After aging, the Cu-SSZ-13 still shows good $deNO_X$ activity, with some changes in the UV–Vis–NIR-DRS spectrum, corresponding to the formation of CuO, though these species must be small as they are not detected using XRD, and potential CuO species are shown in Fig. 3e. Complete APT characterization of samples **6** and **7** is in Fig. 3, Supplementary Figs. 32–34 and

Supplementary Movies 8–10. The NNDs for Al show a small deviation from a random distribution, indicating that aging has caused the removal of some Al from framework positions, and aggregation into Al and Cu-rich regions is evident from the RDFs. Although this does indicate some material destruction, it is less significant than in Cu-ZSM-5, which is reflected in the better stability and activity of the catalyst. The Cu distribution is also heterogeneous, easily shown in the Cu NNDs, and the Cu RDFs show Cu–Cu and Cu–Al affinities, a difference from what was found in the fresh material where no Cu–Al affinity was observed, and indicates some Cu–Al aggregation in the sample. Therefore, even though no significant formation of Cu aluminate domains is indicated in the aged material, a clear affinity of Cu for Al is demonstrated by APT. The Cu was significantly segregated such that Cu-rich regions could be easily identified using isoconcentration surface analysis and compositional histograms. Although there is a less pronounced increase in Al in the Cu-rich regions compared with aged Cu-ZSM-5, it is still present, consistent with limited Cu and Al migration with aging. These observations agree with the known stabilizing influence of Cu in SSZ-13, which have been observed since the earliest reports on the material, and is thought to be caused by the Cu(II) exchanging onto paired Al sites, which stabilizes the material by preventing hydrolysis of the Al–O bonds[10]. Overall, this leads to limited material destruction and explains the remarkable stability of the material under demanding conditions.

## Discussion

The distributions of Cu and Al in the fresh and aged Cu-ZSM-5 and Cu-SSZ-13 materials help to explain the $deNO_X$ activity, shown in Fig. 1, as APT can detect features that are not observable by any other characterization techniques. A comparison of $20 \times 20 \times 20$ nm regions from the four zeolite samples is shown in Fig. 4. Fresh zeolite Cu-SSZ-13 has a random Al distribution, but in fresh zeolite Cu-ZSM-5 the Al distribution is slightly

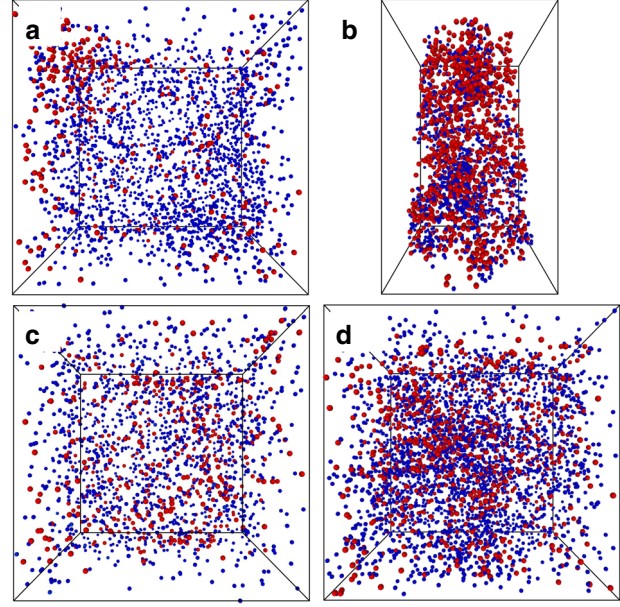

**Fig. 4** Selection of 3D atom probe tomography data for comparison. $20 \times 20 \times 20$ nm³ cubic regions selected from each sample to give a closer comparison of the Cu (red) and Al (blue) distributions, with locations shown in Supplementary Fig. 21, and a schematic overview of these results is given in Fig. 1. **a** Fresh Cu-ZSM-5 (**1**), **b** aged Cu-ZSM-5 (**3**, the bounding box is $20 \times 13 \times 13$ nm³ as this needle was smaller), **c** fresh Cu-SSZ-13 (**4**), and **d** aged Cu-SSZ-13 (**7**)

heterogeneous, likely reflecting the different zeolite synthesis procedures, but highlighting the importance of understanding that real-life catalyst materials may be prepared under non-ideal conditions to satisfy economic requirements, leading to structural and compositional imperfections in the materials that are not encountered under more time consuming, small-scale laboratory preparation procedures, and this has been recently shown by several groups using a variety of techniques[41–45]. Both materials were Cu-exchanged using conventional aqueous procedures, followed by calcination. Previous characterization studies have shown that this results in well-dispersed Cu ions, though it is known to be more difficult to exchange Cu in small-pore materials[50]. However, heterogeneities found in both materials from analysis of the APT data show that Cu-rich regions tend to form. It is possible that the lower activity of the fresh zeolite Cu-ZSM-5 compared to the fresh zeolite Cu-SSZ-13 can be at least partially attributed to differences in the Cu dispersion, but we do not currently have sufficient data to clearly demonstrate this hypothesis. A recent experimental and theoretical report has demonstrated that $NO_X$ SCR with Cu-SSZ-13 falls outside the conventional boundaries of homogeneous or heterogeneous catalysis as the reaction exhibits a density-dependent interaction of multiple ionically tethered single sites[51]. Of importance to the present findings is that optimization of the Cu spatial distribution or mobility is vital to improving low-temperature performance, as Cu ions are found to have a maximum diffusion distance of ~0.9 nm due to electrostatic tethering. Although a random Cu distribution is normally assumed, we have shown in fresh Cu-SSZ-13 that this is not the case, as Cu-rich regions have been identified (Fig. 3), which may have important implications for forming dynamic multinuclear sites, and may also lead for highly heterogeneous reaction behavior within a single zeolite crystal.

After aging according to an industrially relevant methodology, the contrasts between the zeolite frameworks are well-pronounced. In the aged zeolite Cu-SSZ-13, APT shows Cu and Al clustering, but to a limited degree. This is not strongly reflected in the $deNO_X$ activity, and it is possible that this is due to the high conversions as well as the temperatures examined[50]. The changes in the UV–Vis-NIR-DRS spectrum after aging indicate the possible formation of CuO species, which is consistent with the change in the Cu distribution as probed with APT, providing important secondary confirmation as the CuO region is overlapped with LMCT transitions of $Cu^{2+}$. Significant changes for aged zeolite Cu-ZSM-5 were evident from the $deNO_X$ activity, UV–Vis-NIR-DRS spectroscopy as well as APT. The APT iso-concentration surface analysis shows that both the Cu and Al migrate and segregate, and the formation of Cu-rich regions, Al-rich regions, as well as Cu aluminate domains, has been identified. This is an important observation, and one that is not trivial as the paramagnetic Cu complicates $^{27}Al$ NMR data collection and quantification to understand changes to the local environment of Al, which are too small to be detected by XRD. We have demonstrated using APT that as the Al is lost from framework positions it migrates into Al-rich regions, and the formation of Cu aluminate domains is indicated by the affinity of Cu for Al, and vice versa, after aging, leading to Cu–Al aggregation. In aged zeolite Cu-ZSM-5 domains of $CuAl_2O_4$ spinel have been mapped in 3D and their proportion relative to all Cu and Al species in the material has been estimated, giving insight into material deactivation. In the CHA framework the small 8 MR pores are very stable, preventing Al migration and subsequent deactivating Al–Cu clustering to the same extent as is possible in the MFI framework with its larger 10 MR pores, greatly limiting the degradation of this material, as is illustrated in Fig. 1. However, the CHA pores allow Cu migration, as recently demonstrated to be vital for low-temperature reactivity, though as there is limited

migration of Al, the Cu remains electrostatically tethered and therefore catalytically active, explaining the retained performance of the material as inactive copper aluminate species are not significantly formed[51].

In summary, zeolite Cu-SSZ-13 provides superior catalytic performance, and is importantly able to maintain that performance after a simulated 135,000 mile aging, whereas the $deNO_X$ performance of Cu-ZSM-5 degrades significantly. This difference has been ascribed to framework destruction through the removal of Al, which serves as the exchange site for Cu, causing the formation of CuO and $CuAl_2O_4$ species, which have been previously identified with bulk characterization methods. Using APT as a sensitive nanoscale chemical imaging method, we have studied the 3D distribution of Cu, the active site, as well as the inorganic framework elements, in particular Al, O, and Si, in both zeolites Cu-SSZ-13 and Cu-ZSM-5. Both fresh catalyst materials contain a heterogeneous Cu distribution, though to a much greater extent in zeolite Cu-ZSM-5, which also contains a heterogeneous Al distribution. After aging, zeolite Cu-SSZ-13 shows some Cu and Al clustering, whereas zeolite Cu-ZSM-5 shows severe Cu and Al aggregation, and the presence of copper aluminate (i.e., a $CuAl_2O_4$ spinel phase) is demonstrated and mapped in 3D. The result of finding nanometer scale heterogeneities that nicely correlate with catalyst activity is only possible due to the application of APT as a powerful local characterization method. The findings of this study further reinforce the fundamental mechanisms behind the stability of the CHA framework under demanding tailpipe reaction conditions.

## Methods

**Catalyst materials and aging**. The SSZ-13 was prepared in our lab as a commercial version is not available. Hereto, a 25 % solution of the structure directing agent (SDA, *N,N,N*-trimethyl-1-adamantammonium, Sachem) was added to tetraethylorthosilicate (TEOS, Aldrich, >99 %) and aluminum isopropoxide (Acros Organics 98 %+). The resulting mixture was aged at RT (~4 days). After this a 51 % HF solution (Acros organics, 48–51 %) was added and stirred into a homogeneous gel. The gel was transferred into three Teflon lined autoclaves, in equal portions. Autoclaves were sealed and put in a static oven at 150 °C for 6 days. After the synthesis was finished, the resulting solid was washed thoroughly with demineralized water (~8 L). The resulting solid was a white powder. Calcination was performed in a static oven with the following temperature program: heat from room temperature to 150 °C in 2 h 10 min, hold for 1.5 h, then heat to 350 °C over 1.5 h with a 3 h hold, then heat to 580 °C over 4 h 50 min and hold for 3 h and finally cool to room temperature. Crystallinity was evaluated with XRD. The method used has been adapted from Lezcano-Gonzalez and colleagues[3]. The resulting Si/Al ratio was ~20 as determined using ICP-OES. The Cu ion exchange was performed using 1 g SSZ-13 with 50 mL of a 0.1 M $CuSO_4·5H_2O$ (Merck ACS, ISO, Reag. Ph Eur) solution (pH = 4.3) at 80 °C for 2 h. The resulting Cu-SSZ-13 material was washed with demineralized water and dried at 60 °C overnight. Calcination was performed in a static oven with the following temperature program: heat from room temperature to 120 °C in 1 h, hold for 30 min, then heat to 550 °C over 7 h 10 min and hold for 4 h, and finally cool to room temperature.

Zeolite ZSM-5 was purchased from Zeolyst (CBV2314, Lot 2200–89), with a specified Si/Al ratio of 11.5. Before Cu exchange, the ZSM-5 was calcined in air (1 K min$^{-1}$ to 120 °C, hold 30 min, ramp 1 K min$^{-1}$ to 550 °C and hold for 4 h). The Cu ion exchange was conducted following the method of ref.[52]. A 10 mM solution of Cu(II) acetate was used without pH adjustment, with 250 mL solution per gram catalyst, and the suspension was stirred at room temperature for 24 h, and recovered using filtration and washed with a large excess of demineralized water. After drying, the material was recalcined in air using the same program.

Aging was conducted based on the industry standard simulation for a 135,000 mile vehicle-aged catalyst[28]. The catalyst was ramped to 800 °C at 2 K min$^{-1}$ and held for 16 h. The steaming was conducted using 10% steam in air flowing at 100 mL min$^{-1}$, with the steam was created using a bubbler at 47 °C. Steam was first introduced when the catalyst reached 120 °C to prevent condensation. The steaming was stopped prior to cooling the catalyst back to room temperature and was switched to dry air.

**Catalyst testing**. Catalytic activity tests were performed in a fixed bed plug flow set up. Typically, 50 mg of powdered catalyst material (sieve fractions of 0.425–0.150 mm) was loaded in a 1 cm OD quartz tubular reactor. Prior to the experiment, the zeolite sample was pre-treated for 1 h with 5 % $O_2$ in He at 550 °C. After the pre-treatment, the desired reaction temperature was fixed, and then the

catalyst exposed to a SCR feed composition of 1000 ppm NO, 1000 ppm $NH_3$ and 5 % $O_2$, and He for balance, with a Gas Hourly Space Velocity (GHSV) of 100,000 $h^{-1}$. Steady-state measurements were performed at different reaction temperatures, from 150 to 450 °C, using a stabilization period of 60 min at each temperature and analyzing the output gases by mass spectrometry (Hiden Analytical, HPR-20 QIC) and FT-IR gas analysis. All SCR gases were provided by Linde. To avoid condensation in the reaction system, all the gas lines were heated to 150 °C. Equations 1 and 2 were used to calculate respectively the NO conversion and $N_2$ selectivity.

$$\text{NO conversion}(\%) = \frac{\text{NO}_{in} - \text{NO}_{out}}{\text{NO}_{in}} \times 100 \qquad (1)$$

$$\text{N}_2 \text{ selectivity}(\%) = \frac{\text{NO}_{in} - \text{NO}_{out} - \text{N}_2\text{O}_{out} - \text{NO}_{2\,out}}{\text{NO}_{in}} \times 100 \qquad (2)$$

**Catalyst characterization.** UV–Vis-NIR-DRS was collected using a Varian Cary 500 UV–Vis-NIR spectrometer equipped with a DRS accessory to allow collection in the diffuse reflectance mode, against a pure white reference standard. Spectra were collected between 4000 and 50,000 $cm^{-1}$ with a data interval of 10 $cm^{-1}$ and at a rate of 6000 $cm^{-1}$ $min^{-1}$. The UV–Vis-NIR-DRS spectra were corrected for the detector/grating and light source changeover steps at 11,400, 12,500, and 28,570 $cm^{-1}$, respectively. Dehydrated, $O_2$ activated samples were prepared under dry $O_2$ flowing at 30 mL $min^{-1}$ by heating to 450 °C at a rate of 5 K $min^{-1}$, with a 2 h dwell before cooling to room temperature for measurement.

X-ray diffraction (XRD) patterns were recorded on a Bruker D2 X-ray powder diffractometer equipped with a Co $K_\alpha$ X-ray tube ($\lambda = 1.7902$ Å).

The chemical composition of the Cu-SSZ-13 zeolite sample has been determined with Optical Emission Spectroscopy (OES) after activation with Inductively Coupled Plasma (ICP). ICP-OES has been performed by the Geolab (Utrecht University), using a SPECTRO CIROSCCD (by SPECTRO Analytical Instruments GmbH–Germany). Samples were dissolved using an aqua regia with HF solution, in which they were dissolved at 90 °C overnight, after which it was cooled down to RT and neutralized using boric acid. After this the solutions were diluted to yield to appropriate concentrations. The chemical composition of the Cu-ZSM-5 was measured using energy dispersive spectroscopy (EDS) with a EDAX SUTW Sapphire Detector in a Philips XL30 scanning electron microscope (SEM) at an accelerating voltage of 20 kV.

Temperature-programmed desorption of ammonia ($NH_3$-TPD) was performed on a Micromeritics Autochem II 2920 equipped with a TCD detector. Prior to TPD, 0.1 g of catalyst was first out gassed in He for 1 h at 600 °C with a heating ramp of 10 °C $min^{-1}$. Ammonia was adsorbed at 100 °C until saturated, followed by flushing with He for 120 min at 100 °C. The ammonia desorption was monitored using the TCD detector until 600 °C with a ramp of 5 °C $min^{-1}$, using a flow of 25 mL $min^{-1}$.

SEM has been performed on a FEI Nova 200 Dual-Beam SEM/FIB located within the Center for Nanophase Materials Sciences (CNMS) at Oak Ridge National Laboratory (ORNL). The microscope is equipped with the following options: FEG scanning electron microscope, Ion column with Ga liquid ion source for milling, GIS for Pt deposition, Kleindiek nanomanipulator for specimen lift-out and AutoTEM, AutoFIB, and slice and view automation software.

**Atom probe tomography.** Needles for APT analysis were prepared from the four sets of sample under study, namely fresh Cu-ZSM-5 and Cu-SSZ-13 as well as aged Cu-ZSM-5 and Cu-SSZ-13, using FIB milling. For this purpose, seven samples that yielded APT data sets are summarized in Supplementary Table 1. A representative set of SEM images of the APT needles are shown in Supplementary Fig. 6. For Cu-SSZ-13 the crystals were large enough (~4 μm) that standard lift-out and needle preparation were performed using standard specimen preparation techniques utilizing Si micro-tip arrays purchased from CAMECA[53]. For Cu-ZSM-5 the zeolite crystal aggregates were too small for standard preparation, so crystals were attached to the lift-out needle by electrostatic force and transferred to Si micro tips where they were FIB milled into needle-shaped specimens. The needle specimens were transferred to the LEAP 4000XR local electrode atom probe equipped with laser pulsing capabilities and an energy compensating reflectron lens located within the CNMS at ORNL. The specimens were run in laser pulse mode with a laser energy of 200 pJ, base temperature of 40 K, pulse repetition rate of 200 kHz, and a detection rate of 1 atom per 200 pulses. The detector has an efficiency of ~37%. Data analysis is described in the supporting methods, following our previous publication[54]. Videos of the APT sample reconstructions are available in the supporting information to aid with visualizing the ion distributions, clusters and isoconcentration surfaces in 3D.

**Data availability.** All data are available from the corresponding authors on reasonable request.

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

## Acknowledgements

This work is supported by the NWO Gravitation program, Netherlands Center for Multiscale Catalytic Energy Conversion (MCEC), and a European Research Council (ERC) Advanced Grant (No. 321140). The APT measurements were conducted at the Center for Nanophase Materials Sciences, which is a DOE Office of Science User Facility. J.S. has received funding from the European Union's Horizon 2020 research and innovation program under the Marie Sklodowska-Curie Grant Agreement No. 702149. This manuscript has been authored by UT-Battelle, LLC under Contract No. DE-AC05–00OR22725 with the U.S. Department of Energy. The United States Government retains and the publisher, by accepting the article for publication, acknowledges that the United States Government retains a non-exclusive, paid-up, irrevocable, world-wide license to publish or reproduce the published form of this manuscript, or allow others to do so, for United States Government purposes. The Department of Energy will provide public access to these results of federally sponsored research in accordance with the DOE Public Access Plan (http://energy.gov/downloads/doe-public-access-plan).

## Author contributions

J.E.S., R.O., J.D.P., and B.M.W. conceived the project, designed the experiments, and interpreted the data. J.E.S. and R.O. synthesized and characterized the materials. R.O. measured UV–Vis-NIR-DRS spectra, NH$_3$-TPD, and NO$_X$ reduction reactions. J.E.S, W.G., and J.D.P. prepared samples for APT, collected, and analyzed the APT data. All authors participated in the writing of the manuscript.

## Additional information

**Competing interests:** The authors declare no competing financial interests.

