## [Peer Review File · Nature Communications]

Reviewers' comments:

Reviewer #1 (Remarks to the Author):

The authors have applied an innovative and powerful technique, atom probe tomography, to an important class of catalysts used for the ammonia catalytic selective reduction of NO_x. Unfortunately, this turned out to be not the right problem to apply this technique as the information gathered by the analysis is (mostly) similar to the current understanding of the structure and deactivation mechanisms of Cu-containing SCR zeolite catalysts. This can be gathered from the abstract that states that "The application of APT as a sensitive and local characterization method confirms previously theorized deactivation mechanisms with the quantitative identification of nanometer scale heterogeneities". Since in the end there are no substantial improvements in our understanding of the function and deactivation of the Cu-zeolite catalysts, I do not recommend this report for publication in Nature Communications. I disagree strongly with 'previously theorized deactivation mechanisms' in the authors sentence above, as on the basis of solid-state NMR spectroscopy, X-ray spectroscopy, EPR spectroscopy and adsorption investigations, the differences between the deactivation of ZSM-5 and SSZ-13 were pretty clear and largely consistent with the observations reported here.

I welcome the level of structural and computational detail that can be obtained from the APT technique, but in this case the findings are underwhelming. Perhaps the nicest piece of information in the report is the appearance of a correlation between Cu and Al in the SSZ-13 aged samples. This can be explained by the known fact that Cu(II) ion exchange increases the stability of the Al atoms associated with the Cu. The authors show that those Al atoms not associated with Cu (the acid sites) are less stable to the aging protocol than the Cu-exchange ones (as was already recognized).

In addition the paper could improve by using classical catalytic materials nomenclature. For instance, the authors talk about Cu-Al affinity (line 191) on the aged samples. This will be more clearly expressed as Cu-Al aggregation in the sample. Terms, such as 'weak affinity' (line 201) obfuscate rather than clarify what has been the effect of aging.

At several points the authors speculate to explain some experimental observations. For instance, it is stated that the 'heterogeneous Cu distribution is possibly caused by preferential material accessibility during the exchange' (lines 92-93). I don't know the origin of the heterogeneous Cu distribution, but it is certainly not accessibility since in SSZ-13, with smaller pores, there are no accessibility problems. The lack of correlation between Al and Cu in this sample strongly suggest to this reviewer that there are artifacts in this technique (or data treatment) of which the authors are unaware. Because differences between (purportedly homogeneous) samples were observed, this is assigned to 'the imperfect nature of industrial zeolite crystals'. But the use of two sub-micron sized samples to make inferences about bulk catalysts is really unwarranted. They may have just been unlucky. If such large compositional variability were widespread across the sample, it would be noticed in peak shapes of X-ray diffraction patterns, in solid-state Al and Si NMR spectra, in catalytic properties and other macroscopic processes.

Reviewer #2 (Remarks to the Author):

This manuscript reports the results of an atom probe tomography study of Cu - Al distributions in Cu-exchanged chabazite, a zeolitic material that is used in automotive catalytic converters. Aging and degradation of this material, in use, has been associated with Cu-Al mobilisation and segregation, at the nanometer length scale. This paper is novel and of interest because it provides direct evidence of this process, and furthermore identifies the copper aluminate spinel phase that results from the

changes that occur in the chabazite during aging and degradation. The use of atom probe tomography seems particularly effective in this set of experiments, which demonstrate the applicability of the method to these sorts of problems. As such, it is a valuable and novel contribution that marks an example of the utility of this method.

Reviewer #3 (Remarks to the Author):

The authors employ atom probe tomography (APT) to generate 3D, nanoscale reconstructions of two different Cu-exchanged zeolite catalyst materials in both the fresh and aged condition to illuminate the origins of performance decay as an automotive catalyst in one material over another. The APT reconstructions revealed that both pristine materials exhibit a heterogeneous Cu distribution, which was not observable previously. After testing, one material (Cu-ZSM-5) showed degraded performance and APT revealed strong Cu and Al clustering to the point of forming nanoscopic CuAl_2O_4 particles within the zeolite matrix. Conversely, Cu-SSZ-13 showed more limited Cu and Al clustering after test and no significant degradation in catalyst activity. Taken at face value, the characterization and potential impact of these results are compelling. The application of APT to this materials system is novel and challenging, even for experienced practitioners, and the authors deserve credit for collecting such an interesting set of data and presenting a strong story. I further believe that the results will be interesting to a broad readership base. That being said, I do have technical questions regarding the APT data and its interpretation that must be addressed prior to making a final decision of this works publication. (Daniel Schreiber - PNNL)

Major Issues:

1. Dataset Aged Cu-ZSM-5 (Sample 3) is the linchpin for the conclusions on degraded performance in aged Cu-ZSM-5, but it is also a troubling dataset from an objective APT perspective.

1a. The dimensions of this reconstruction suggest that the tip fractured very early, probably within a few hundred thousand collected ions? Some in the APT community argue that we should discard the first couple million ions to avoid FIB artifacts, which likely includes this entire reconstructed volume several times over. The authors need to provide some justification for the validity and integrity of this seemingly questionable dataset.

1ai. What is the Ga concentration/distribution? How does it compare to the other datasets? This near-surface damage is always present in FIB-prepared specimens. What gives you sufficient confidence in this particular dataset?

1a.ii. Was the evaporation smooth and controlled throughout?

1a.iii. Do any of the other datasets from the same material/condition (however small) give support to the key observations of this dataset (particularly the strong Al-Cu clustering and possible CuAl_2O_4 formation)?

1b. A robust sister dataset from the same material condition would be ideal to support the validity of this observation. Presumably if the data existed the authors would have included it already. Could the authors provide some perspective on how much effort went into collecting this single dataset and the feasibility of collecting more data?

1c. Page 3, line 112-114 / Fig S14: "The proportion of the material contained in these Cu aluminate regions was quantitatively assessed from the APT data and found to include 20% of all Cu and 35% of all Al."

1ci. How was this quantification made? Atom counts or volume fraction? If atom counts, did you subtract the matrix contribution for Al and Cu?

1cii. Is the aged matrix quantitatively depleted of Al and Cu as a result (normalized to Si presumably)?

1ciii. Considering the very small size of this dataset, how can this observation be reliably extrapolated to the overall prevalence of Cu aluminate regions throughout the bulk material?

2. Details on Cluster Analysis:

2a. The method described in the supplemental information for optimization of d_{max} , N_{Min} and order is OK, but the plots of the optimization process (or a sensitivity analysis by another term) must be presented for each dataset. This will provide critical details both for replication of these results and also justify the validity of the cluster identification.

2b. Details should also be given for how you are treating molecular ions (e.g. AlO^+) and elemental peaks (e.g. Al^{2+}) when performing the Al cluster search.

3. Compositional measurements (Table S2): There are some peculiarities in this table that should be addressed.

3a. Why was background correction not performed? This can have a profound effect on the composition measurements, particularly for repeatability from tip-to-tip

3b. The O concentration varies significantly amongst the datasets. Please discuss the origins of these variations, particularly the high O concentration for the Fresh Cu-SSZ-13 (~73 at.%). Are these variabilities APT artifacts or reflect something within the material itself?

3c. Aging increased the measured concentrations of Al and Cu in both materials, both in absolute concentration and when normalized against Si. Are there Al/Cu-poor regions outside the APT field of view? Was Si selectively lost during the aging test? Again please discuss the origin of these variations to increase the confidence in the APT measurements.

3d. Along the same line, the ratio of Al:Si increases by about a factor of 2 upon aging for both materials. However, the Cu:Si ratio increases by a factor of 4 for Cu-ZSM-5 and only a factor of ~1.4 for Cu-SSZ-13. Based only on that measurement it would appear that Cu is behaving more differently between the two materials than Al. Could the authors comment on this difference and whether or not they believe it is real and significant.

3e. It would be helpful to add a column to differentiate "matrix" and "bulk" measurements in this summary Table S2.

4. RDF plots:

4a. How are molecular ions handled when calculating the RDF profiles?

4b. In section S5 the authors state: "Once clusters are determined, the RDF can be used to examine compositional heterogeneities that exist within clusters..." This statement implies that you are only performing the RDF from the volume of material defined by a previous cluster search output. This would create a huge bias that in the resulting RDF plots that was not discussed directly in the text. I do not believe that is actually what happened but this point must be clarified. If the RDF is only performed on the cluster search output, the authors must justify their logic and the meaning of a

"normalized composition" in that context.

Minor Issues:

1. Abstract and page 2 line 46: "...determined with sub-nanometer resolution using atom probe tomography..."; "...were mapped with sub-nm resolution..."

This is an unrealistic spatial resolution for a heterogeneous oxide, as shown by your proximity histograms where a strong matrix signal extends throughout the $\sim\text{CuAl}_2\text{O}_4$ particle (Fig S15). Please rephrase these statements.

2. Page 2, line 72-75 + Ref 39: Mapping small concentrations of Fe within a silicate network is very challenging for APT due to the overlap of the primary Fe peak at $^{56}\text{Fe}^{2+}$ with the more dominant $^{28}\text{Si}^{1+}$. In the current paper you may want to highlight that such convolutions do not exist for a Cu exchanged zeolite, similar to the Al zeolite in Ref 40 that did show Al clustering.

3. Figures 2 and 3: Using "normalized concentration" in the concentration profiles / proximity histograms obscures potentially informative data, including potential flaws and anomalies. These should be drawn with more common units of at.%, at least in the supplemental information. RDFs are OK as they are natively normalized.

4. Figure 4: Subjective opinion, but I think these images would be more powerful if you use color-indexed cluster search output rather than raw atom maps.

5. Throughout: Please make clear that the concentrations reported are in units of at.%, or clarify when that is not the case

6. Page 3 and elsewhere: "isosurface" should be replaced by "isoconcentration surface" which is the more precise term

7. Section S5: Add representative mass spectra from all four materials/conditions. Log-scale preferable. If the ranging/peak identifications are different please also include that information.

8. Section S5: Isosurface analysis: Clearly define your selected values for voxel size and delocalization so that others can replicate your results.

Point-by-Point Responses to the Referee's Comments - NCOMMS-17-11574-T

We sincerely thank the three referees for their very valuable and constructive input on our paper as well as the related suggestions for revisions and additional experimental data/technical explanations, which are now included in the revised manuscript.

Below, you will find:

- in black - the comments of the reviewers
- in blue - our replies to the reviewers comments

Reviewer #1:

The authors have applied an innovative and powerful technique, atom probe tomography, to an important class of catalysts used for the ammonia catalytic selective reduction of NO_x. Unfortunately, this turned out to be not the right problem to apply this technique as the information gathered by the analysis is (mostly) similar to the current understanding of the structure and deactivation mechanisms of Cu-containing SCR zeolite catalysts. This can be gathered from the abstract that states that "The application of APT as a sensitive and local characterization method confirms previously theorized deactivation mechanisms with the quantitative identification of nanometer scale heterogeneities". Since in the end there are no substantial improvements in our understanding of the function and deactivation of the Cu-zeolite catalysts, I do not recommend this report for publication in Nature Communications. I disagree strongly with 'previously theorized deactivation mechanisms' in the authors sentence above, as on the basis of solid-state NMR spectroscopy, X-ray spectroscopy, EPR spectroscopy and adsorption investigations, the differences between the deactivation of ZSM-5 and SSZ-13 were pretty clear and largely consistent with the observations reported here.

Response

We thank the referee for his/her frank and upfront review, and for stating the importance of studying this class of catalyst materials. We would like to point out that details of deactivation mechanisms of solid catalysts are in fact very difficult to detect by e.g. solid-state NMR spectroscopy and X-ray spectroscopy due to the fact that there are local heterogeneities that exist at the nanoscale. X-ray diffraction cannot detect many of these deactivating species due to the very small size of the domains, while NMR cannot be always reliably applied to these materials as paramagnetic copper may interfere with the signal. Consequently, we have modified the introduction to reinforce this point by adding: "However, there still remains significant gaps in our understanding of the exact deactivation mechanisms in these materials due to the small size of deactivating species, which makes them invisible to many techniques, e.g. X-ray diffraction, and the presence of paramagnetic Cu²⁺ can interfere with locally sensitive techniques, e.g. NMR. Therefore, it is necessary to reconstruct the distribution of single atoms in 3D to gain a more complete picture of deactivation." We have also modified the last sentence of the abstract: "The application of APT as a sensitive and local characterization method provides quantitative identification of nanometer scale heterogeneities that lead to material deactivation."

The application of APT to this system is indeed novel and challenging. As the referee states the findings are "(mostly) similar" to the current understanding. One could argue that this statement in itself underscores the need for such type of nano-scale investigations, which are very much needed as even incremental increases in our understanding creates substantial advances in the related automotive catalytic technology. We believe that the insights reported in this manuscript are novel and worthy of publication in a journal, such as *Nature Communications*. This assertion is supported by reviewers 2 and 3 in some of the following comments: "As such, it is a valuable and novel contribution that marks an example of the utility of this method." and "The application of APT to this materials system is novel and challenging, even for experienced practitioners, and the authors deserve credit for collecting such an interesting set of data and presenting a strong story. I further believe that the results will be interesting to a broad readership base."

Furthermore, while this work has been discounted "as the information gathered by the analysis is (mostly) similar to the current understanding of the structure and deactivation mechanisms" we believe that there is great value in these types of nanoscale investigations. This assertion is certainly supported by the larger research community as the work of our group was recently featured as a *Chemical & Engineering News* cover story (issue of July 17, 2017), "Hunting for the hidden chemistry in heterogeneous catalysts," and this article features the use of APT to characterize solid catalysts, along with a number of other advanced characterization techniques.

Response

We thank the referee for highlighting the importance of this observation in the manuscript, and we have highlighted this by adding the following statement: "These observations are consistent with the known stabilizing influence of Cu in SSZ-13, which have been observed since the earliest reports on the material, and should be caused by the Cu(II) exchanging onto paired Al sites and prevent material destruction." We however do not believe it is underwhelming due to the difficulty in detecting these influences for the aforementioned reasons of the sensitivity of X-ray diffraction to small domain sizes and paramagnetic Cu disrupting NMR quantification.

In addition the paper could improve by using classical catalytic materials nomenclature. For instance, the authors talk about Cu-Al affinity (line 191) on the aged samples. This will be more clearly expressed as Cu-Al aggregation in the sample. Terms, such as 'weak affinity' (line 201) obfuscate rather than clarify what has been the effect of aging.

Response

We thank the referee for pointing out that we were not being clear in our terminology to a broader audience. We have modified the language of the manuscript to reflect this comment and believe that the terminology is now more clear with respect to the results of aging, and consistent with the nomenclature used in the field.

At several points the authors speculate to explain some experimental observations. For instance, it is stated that the 'heterogeneous Cu distribution is possibly caused by preferential material accessibility during the exchange' (lines 92-93). I don't know the origin

of the heterogeneous Cu distribution, but it is certainly not accessibility since in SSZ-13, with smaller pores, there are no accessibility problems.

Response

Examining the recent literature shows that it is well known that it can be unpredictable and much more difficult to exchange copper into SSZ-13, due to its small pores, than into a larger pore framework such as MFI or BEA (for example see doi: 10.1039/C4CY00384E, 10.1021/acscatal.5b01200). This immediately points to a potential for nanoscale heterogeneities. In line with this, we have clarified the statement mentioned above in the manuscript to reinforce the somewhat unpredictable results of aqueous ion exchanges, especially in small pore zeolites, and believe this has improved the manuscript. "It is known in small pore zeolites that conventional aqueous ion exchange of Cu can be unpredictable and more challenging than larger pore materials and this non-random Cu dispersion may be the result of this known challenge.^{43,44} This finding shows the Cu dispersion challenges the notion that these materials have a homogeneous Cu dispersion, as it can clearly be heterogeneous, though still as spectroscopically isolated Cu, giving a highly active catalyst. Additionally, it highlights the power of APT to probe these materials as no other techniques can resolve isolated Cu ions in 3D^{28,30,33,34,37,38}." While the possibility of an inhomogeneous copper exchange has been quickly rejected, this has been reported before, though at longer length scales than we investigate (for example see work from the group of Prof. Corma, doi: 10.1021/cs400499p).

The lack of correlation between Al and Cu in this sample strongly suggest to this reviewer that there are artifacts in this technique (or data treatment) of which the authors are unaware.

Response

While APT does have the highest spatial resolution of any 3D tomography technique, there are still limits to what it can spatially resolve, and all atomic distances will be blurred due to spatial noise from the technique. With a rough calculation from the Si/Al ratio, the density of Al atoms is such that they will be spaced in 3D closer than the resolution of the technique. This calculation, along with the high amount of Cu, means that both atoms will be on average closer than the resolution of the technique, such that a correlation cannot be found, within the resolution of APT. However, we do state in the manuscript "The RDF for Al shows an Al-Cu affinity, though this would be expected as Al serves as the exchange site for Cu, shown schematically in Fig. 3b, highlighting the sensitivity of APT." This statement (which is supported by the data presented) shows there is in fact a correlation, unlike this point of the reviewer.

Because differences between (purportedly homogeneous) samples were observed, this is assigned to 'the imperfect nature of industrial zeolite crystals'. But the use of two sub-micron sized samples to make inferences about bulk catalysts is really unwarranted. They may have just been unlucky. If such large compositional variability were widespread across the sample, it would be noticed in peak shapes of X-ray diffraction patterns, in solid-state Al and Si NMR spectra, in catalytic properties and other macroscopic processes.

Response

There are a number of recent manuscripts highlighting “large compositional variability” that is missed by bulk characterization methods, as they will give an ensemble average. This has only recently been demonstrated by novel microscale characterization techniques, which really are at the cutting edge of the spatial resolution that is possible. As methods have greatly improved with regards to sensitivity and resolution, there are many new reports showing that with zeolites the bulk really is an average of a highly diverse population. Many groups have demonstrated this including: the group of Maarten Roeffaers with industrial mordenite crystals (doi: 10.1021/nn505576p, 10.1021/acscatal.7b01148, 10.1002/cctc.201500708) and the group of Jeroen van Bokhoven on industrial MFI (doi: 10.1002/chem.201406182) and any one of a large number of fine publications from the group of Javier Pérez-Ramírez (for example doi: 10.1039/C7MH00088J, 10.1002/adfm.201601748).

The examples by these groups (and of course other scientists we have missed in this very short recounting) demonstrate that the complexity of zeolite catalysts is even more than was previously considered, as microscale techniques are revealing that what we find using bulk characterizations is merely an ensemble average of a highly heterogeneous population on the micro/nano-scale. Further, as most publications we mention here are very recent, we believe this will be only further reinforced with future investigations, which are at the forefront of technical capabilities due to the small size and complex nature of industrial zeolite catalysts. We have added the following statement to the manuscript to clarify this point: “Additionally, the recent application of advanced micro- and nanoscale characterization techniques has revealed that small, industrial zeolite crystals can be quite heterogeneous, and that the results of bulk studies give an ensemble average from a diverse population, supporting the present findings.^{41–45}”

Reviewer #2:

This manuscript reports the results of an atom probe tomography study of Cu - Al distributions in Cu-exchanged chabazite, a zeolitic material that is used in automotive catalytic converters. Aging and degradation of this material, in use, has been associated with Cu-Al mobilisation and segregation, at the nanometer length scale. This paper is novel and of interest because it provides direct evidence of this process, and furthermore identifies the copper aluminate spinel phase that results from the changes that occur in the chabazite during aging and degradation. The use of atom probe tomography seems particularly effective in this set of experiments, which demonstrate the applicability of the method to these sorts of problems. As such, it is a valuable and novel contribution that marks an example of the utility of this method.

Response

We would like to sincerely thank reviewer 2 for his/her supportive comments and believe these offer a strong rebuttal to the comments of reviewer 1, and demonstrate this manuscript is a candidate for a scientific journal, such as *Nature Communications*.

Reviewer #3:

The authors employ atom probe tomography (APT) to generate 3D, nanoscale reconstructions of two different Cu-exchanged zeolite catalyst materials in both the fresh and aged condition to illuminate the origins of performance decay as an automotive catalyst in one material over another. The APT reconstructions revealed that both pristine materials exhibit a heterogeneous Cu distribution, which was not observable previously. After testing, one material (Cu-ZSM-5) showed degraded performance and APT revealed strong Cu and Al clustering to the point of forming nanoscopic CuAl_2O_4 particles within the zeolite matrix. Conversely, Cu-SSZ-13 showed more limited Cu and Al clustering after test and no significant degradation in catalyst activity. Taken at face value, the characterization and potential impact of these results are compelling. The application of APT to this materials system is novel and challenging, even for experienced practitioners, and the authors deserve credit for collecting such an interesting set of data and presenting a strong story. I further believe that the results will be interesting to a broad readership base. That being said, I do have technical questions regarding the APT data and its interpretation that must be addressed prior to making a final decision of this works publication.

Response

We would like to thank reviewer 3 for his/her assessment and for the technical questions. We have addressed these in a significantly revised form of the manuscript and the Supporting Information, and have also provided a point-by-point responses to the comments below. We would also like to highlight that we interpret these comments not as a devaluation of the impact of the manuscript, but that they have been made to ensure a complete and rigorous report of the details of the analysis, and as such have revised the manuscript and Supporting Information.

Major Issues:

1. Dataset Aged Cu-ZSM-5 (Sample 3) is the linchpin for the conclusions on degraded performance in aged Cu-ZSM-5, but it is also a troubling dataset from an objective APT perspective.

Response

We will provide point-by-point responses to each individual comment below. However, as a general response we have already stated in the manuscript “Two needles of fresh Cu-ZSM-5 and one of aged Cu-ZSM-5 were successfully reconstructed as this material was especially prone to failure” and this statement really understates the technical difficulty of this project. The work presented here was the culmination of three weeks of rigorous/frustrating sample preparation resulting in only a few valid APT datasets due to the low APT experimental yield for the small particles. Most likely, the referee is very well aware of the technical challenges associated with APT, and more specifically with non-conductive materials, which is further exacerbated with small crystal zeolites that consist of complex crystalline intergrowths,

increasing the challenge of specimen preparation and chances of specimen failure. We state in the Supporting Information: “The non-conductive nature of the materials used in this study complicates the data collection. The pulsed laser heating helps to overcome these difficulties. Sample heating creates issues with processing the sample mass spectra as it creates thermal tails in the data. Due to this issue, not all collected data sets gave reliable, quantitative results, and only data sets that gave reliable results are presented.” We found in practice that a high percentage of APT runs gave data, which was not reliable due to early specimen fracture, and sub par mass resolution as well as for the aforementioned reasons. We were very careful to present only the most reliable results.

While we of course wish we could have more data sets, we are confident in the collected data, which is statistically significant. A combination of the confidence in the data and time management lead us to this conclusion. Therefore, we have been forced to work within the constraints of reliable datasets. We have included in the Supporting Information one additional sample of aged Cu-ZSM-5, which we chose not to include in the manuscript as it contained significant Ga implantation. However, the results of this needle are in line with those obtained from the data set currently in the manuscript, so we added them to a separate section of the Supporting Information to reinforce that our conclusions are valid.

1a. The dimensions of this reconstruction suggest that the tip fractured very early, probably within a few hundred thousand collected ions? Some in the APT community argue that we should discard the first couple million ions to avoid FIB artifacts, which likely includes this entire reconstructed volume several times over. The authors need to provide some justification for the validity and integrity of this seemingly questionable dataset.

Response

This is a very good point, and we agree that Ga damage can easily influence the data. You are correct that the dataset is very small, but we were very careful regarding the existence of Ga contamination/damage. We were very careful in the sample preparation to leave minimal Ga damage on the surface. The zeolites were coated with Pt using e-beam deposition. A 30 kV beam was used for the initial milling, leaving an ~250 nm wide needle with several 100 nm's of Pt remaining on the surface. A 2 kV final milling step was used to remove the Pt cap. The zeolite material mills much faster than the Pt cap, and therefore, the tip shape was finally a very sharp needle with a smaller shank angle. We were also careful to minimize the e-beam exposure to the material because zeolites also suffer from damage resulting from the e-beam. Ga has been an issue that we were the most concerned with when selecting the validity of the datasets, and in our analyses such as the RDF, proximity histograms, etc., we extracted the Ga damaged regions from the dataset using a Ga isosurface and exporting the low gradient side of the Ga interface. The figures show the full datasets for aesthetic purposes. We have added a section in the supplementary materials describing this and point the reader in the text here. In the version of the manuscript we had submitted we did not emphasize sufficiently how we dealt with this potential issue, but have now clarified the point: “In the Supporting Information we also include a discussion of potential Ga damage to the materials and how we assured this was not leading to our observations.” To this end the Supporting information now contains a significant section on assessing Ga damage. We have also added the mass spectrum for the four main samples in the manuscript for the region without Ga extracted and with the Ga extracted using an isosurface. We also

included bulk compositions before and after removal of the Ga iso-surface for each sample. You can find all this information in the newly adapted Supplementary Information.

1ai. What is the Ga concentration/distribution? How does it compare to the other datasets? This near-surface damage is always present in FIB-prepared specimens. What gives you sufficient confidence in this particular dataset?

Response

This has been addressed in the revised supplementary materials section. We are confident in this dataset because there is little to no Ga contamination in the mass spectrum after removal of the Ga rich region, and we also have regions with high Ga contamination in the non-steamed SSZ-13 and ZSM-5 materials that show no correlation to Ga and Al or Cu. All of this has been explained in the “Assessing Ga damage” section of the Supporting Information.

1aai. Was the evaporation smooth and controlled throughout?

Response

Running zeolites in APT requires judicious operator oversight as opposed to, i.e. metals. However, when running at low evaporation rates being able to achieve controlled evaporation until the point of sample failure was a stipulation for determining the validity of the data. Below is the detector event histogram from the non-steamed ZSM-5 sample. All other detector event histograms were similar. The white spot is a dead spot on our detector. The high-density spots close to the aperture were not used in the reconstruction.

1aiii. Do any of the other datasets from the same material/condition (however small) give support to the key observations of this dataset (particularly the strong Al-Cu clustering and possible CuAl_2O_4 formation)?

Response

As mentioned before we do have one additional data set for the aged Cu-ZSM-5, but it contains significant Ga implantation so we previously chose not to present it. However, in light of the reviewer's comments, we have now added it to the Supporting Information only. It contains the same general trends of Al and Cu aggregation, but the region with the highest concentration of Cu and Al is at the tip of the needle, with a high Ga concentration as well, therefore we did not feel confident in presenting this data. However, as discussed in the Supporting Information we can preclude this Cu and Al aggregation as being caused by the Ga ion beam due to the other data sets, and instead we are confident it was simply a coincidence to find the aggregation at the tip of the needle. In fact, there is a Al-Cu affinity within the non Ga implanted region, but this is only a very small number of ions.

1b. A robust sister dataset from the same material condition would be ideal to support the validity of this observation. Presumably if the data existed the authors would have included it already. Could the authors provide some perspective on how much effort went into collecting this single dataset and the feasibility of collecting more data?

Response

This has been previously addressed in the general response to 1, but to reiterate the results presented are the culmination of a full 3 weeks of user time at the APT facility at ORNL. We have now added one additional dataset that supports the aged Cu-ZSM-5 results to the Supporting Information.

1c. Page 3, line 112-114 / Fig S14: "The proportion of the material contained in these Cu aluminate regions was quantitatively assessed from the APT data and found to include 20% of all Cu and 35% of all Al."

1ci. How was this quantification made? Atom counts or volume fraction? If atom counts, did you subtract the matrix contribution for Al and Cu?

Response

The quantification was made with atom counts. We did not subtract the matrix contribution as the caption for Figure S14 states "This region was determined to contain ~20 % of all Cu ions and ~35% of all Al ions present in the needle."

1cii. Is the aged matrix quantitatively depleted of Al and Cu as a result (normalized to Si presumably)?

Response

This is indeed the case and can be seen in the proximity histograms in Figure S15 by examining the regions outside of the iso-surfaces.

1ciii. Considering the very small size of this dataset, how can this observation be reliably extrapolated to the overall prevalence of Cu aluminate regions throughout the bulk material?

Response

As stated in response to point 1aiii, the reason we were confident in publishing this analysis is that it agrees with previously reported bulk analyses. While we know APT can have issues with compositional accuracy, the agreement with other results makes us confident this can be copper aluminate. Additionally, we have removed the term “quantitative” from the manuscript to not overemphasize this. The formation of this phase is known based on bulk characterization, but for the first time we are able to localize it in 3D at the nanoscale. We state in the text: “Chemically or spatially resolved insights regarding the deactivation of Cu-exchanged zeolites have been gained using X-ray absorption fine structure (XAFS), transmission electron microscopy (TEM), scanning TEM (STEM), STEM electron dispersion spectroscopy (STEM-EDS) and X-ray photoelectron spectroscopy (XPS), but these techniques suffer from significant drawbacks including the inability to detect isolated ions, as well as offering only 2D information or bulk averages^{14,24,28,30–34,37,38}.” And “Across the 8% Al isoconcentration surface the Al/Cu ratio increases to approximately 2, which matches the stoichiometry of Cu aluminate, a CuAl_2O_4 spinel species, which has long been regarded as one of the species that forms upon aging of these materials. Previously it was only identified by bulk analysis, and here is shown as spatially isolated nanoscale features in 3D^{14,24,31–34}.”

2. Details on Cluster Analysis:

2a. The method described in the supplemental information for optimization of dmax, NMin and order is OK, but the plots of the optimization process (or a sensitivity analysis by another term) must be presented for each dataset. This will provide critical details both for replication of these results and also justify the validity of the cluster identification.

Response

We have updated the Supporting Information to reflect this with additional parameters for each cluster analysis including d-max, Order, N-min, L, d-erosion and cluster count.

2b. Details should also be given for how you are treating molecular ions (e.g. AlO^+) and elemental peaks (e.g. Al^{2+}) when performing the Al cluster search.

Response

We used all Al related peaks for all Al cluster analysis. There were only metallic Cu peaks. All Cu peaks were used in the Cu cluster analysis. This description has been added to the Supplementary Information.

3. Compositional measurements (Table S2): There are some peculiarities in this table that should be addressed.

Response

We will respond to each point separately about this.

3a. Why was background correction not performed? This can have a profound effect on the composition measurements, particularly for repeatability from tip-to-tip.

Response

These are the background corrected values that are presented in the composition table. Due to the oxide nature of these materials, we are already skeptical of the absolute quantification from sample to sample, and we have made this very clear in the manuscript. We are most concerned with trends in the data such as clustering of elements. The compositions before and after removal of the Ga-rich regions have been included in the Supporting Information to be as transparent as possible about the data that was collected so the reader can assess the experimental results. We have updated the Supporting Information to be very clear about this.

3b. The O concentration varies significantly amongst the datasets. Please discuss the origins of these variations, particularly the high O concentration for the Fresh Cu-SSZ-13 (~73 at.%). Are these variabilities APT artifacts or reflect something within the material itself?

Response

We are convinced that the variations in oxygen content are an artifact of the technique due to the complex evaporation of oxygen and the metal-oxide molecules that also evaporate from these materials. One of the issues is the ability for oxygen or oxides molecules to evaporate as double detector hits. If this occurs at the same time in the same region of the detector, oxygen loss can occur. There has also been some work done that shows the existence of molecular ion disassociation after field evaporation, in which one ion becomes deionized and does not hit the detector within the appropriate time-of-flight window and is registered as background. These issues can cause deviations from the actual composition, and hard to control experimental parameters can influence the severity of these artefacts. We have included the compositional data to be completely transparent about these issues and have explained this in more detail in the Supplementary Information (with references) to guide the reader to understand the phenomenon better:

“It is known that there are difficulties in quantitatively detecting oxygen or oxides as these tend to field evaporate as double detector hits, which will be registered as a single hit, and there is also evidence that molecular ions can dissociate after field evaporation, such that one ion becomes deionized and does not hit the detector within the appropriate time-of-flight window and is registered as background (even though we attempted to keep the detection rate low, 1 ion per 200 pulses).⁹⁻¹¹ These issues can cause deviations from the actual composition, and hard to control experimental parameters can influence the severity of these artefacts. Therefore, we have concluded that APT has difficulty in quantifying the Al content in zeolite materials and we emphasized trends (isosurfaces, clustering, radial distribution function, etc.) rather than comparing absolute numbers.”

3c. Aging increased the measured concentrations of Al and Cu in both materials, both in

absolute concentration and when normalized against Si. Are there Al/Cu-poor regions outside the APT field of view? Was Si selectively lost during the aging test? Again please discuss the origin of these variations to increase the confidence in the APT measurements.

Response

As previously mentioned in our responses, and also was discussed in two other APT papers on zeolites, (doi: 10.1002/anie.201606099, 10.1038/ncomms8589) quantification of the Si/Al ratio, etc., can be challenging in these materials. This is why we avoided strict numerical comparisons and instead looked at trends between materials and the important features such as finding migration and aggregation of Cu and Al after aging. During the aging test Si should not be “lost” from the material, but all elements can definitely migrate within a crystal, and as such may have been lost in the limited APT sample volume, but the trends in Cu and Al migration and aggregation we found are all consistent between materials and the results of other studies.

3d. Along the same line, the ratio of Al:Si increases by about a factor of 2 upon aging for both materials. However, the Cu:Si ratio increases by a factor of 4 for Cu-ZSM-5 and only a factor of ~1.4 for Cu-SSZ-13. Based only on that measurement it would appear that Cu is behaving more differently between the two materials than Al. Could the authors comment on this difference and whether or not they believe it is real and significant.

Response

We completely agree with the assessment that Cu is behaving more differently between the materials. This in fact is one of the important reasons why Cu-SSZ-13 is more stable, and the exact reasons are not understood, but this manuscript is an attempt, in a long line of investigations, to add another piece to solving this puzzle. However, in line with our response to 3c we do not want to emphasize the strict numerical comparison. See the response to 3b as well.

3e. It would be helpful to add a column to differentiate “matrix” and “bulk” measurements in this summary Table S2.

We have now added the bulk composition before and after removal of the Ga rich regions, which had only a small influence on the detected composition of each needle.

4. RDF plots:

4a. How are molecular ions handled when calculating the RDF profiles?

Response

All molecular ions are decomposed and we have now added this to the Supporting Information.

4b. In section S5 the authors state: “Once clusters are determined, the RDF can be used to

examine compositional heterogeneities that exist within clusters...” This statement implies that you are only performing the RDF from the volume of material defined by a previous cluster search output. This would create a huge bias that in the resulting RDF plots that was not discussed directly in the text. I do not believe that is actually what happened but this point must be clarified. If the RDF is only performed on the cluster search output, the authors must justify their logic and the meaning of a “normalized composition” in that context.

Response

Thank you for pointing this out so that we can have the most clear explanation in the manuscript. As you state we did not actually do this. We have corrected the terminology to clarify this point and review any ambiguity or misstated implications of the method used.

Minor Issues:

1. Abstract and page 2 line 46: “...determined with sub-nanometer resolution using atom probe tomography...”; “...were mapped with sub-nm resolution...” This is an unrealistic spatial resolution for a heterogeneous oxide, as shown by your proximity histograms where a strong matrix signal extends throughout the ~CuAl₂O₄ particle (Fig S15). Please rephrase these statements.

Response

This is a good point. We have rephrased this from “sub-nm resolution” to “nanometer resolution.” Of course the exact determination of the spatial resolution is quite difficult and can easily change within the reconstruction itself. We have used sub-nm resolution because we have obtained almost angstrom resolution in other oxide materials (not the current zeolites) by imaging lattice planes.

2. Page 2, line 72-75 + Ref 39: Mapping small concentrations of Fe within a silicate network is very challenging for APT due to the overlap of the primary Fe peak at 56Fe²⁺ with the more dominant 28Si¹⁺. In the current paper you may want to highlight that such convolutions do not exist for a Cu exchanged zeolite, similar to the Al zeolite in Ref 40 that did show Al clustering.

Response

Thank you for pointing this out. We have added a sentence to reinforce this point: “A potential challenge in this work is that there is overlap between the primary Fe peak (56Fe²⁺) and the dominant Si peak (28Si⁺) in the mass spectrum, which could complicate the quantification and subsequent analysis, but no similar overlap is encountered in this work for Cu.”

3. Figures 2 and 3: Using “normalized concentration” in the concentration profiles / proximity histograms obscures potentially informative data, including potential flaws and anomalies. These should be drawn with more common units of at.%, at least in the supplemental information. RDFs are OK as they are natively normalized.

Response

We prefer to keep the data presented in normalized concentration because of the variation in the oxide composition from sample to sample, and we are mostly concerned about the Al-Al, Cu-Cu, and Al-Cu affinities, and not the exact atomic concentration. This also reiterates our point that the exact atomic concentrations for these oxide samples are not very accurate using APT. If we were to add atomic percent, the reader can be confused about the data. We have added the results in the Supporting Information with atomic percent in the section on assessing Ga damage as it is useful to see just how low the Ga content is.

4. Figure 4: Subjective opinion, but I think these images would be more powerful if you use color-indexed cluster search output rather than raw atom maps.

Response

We prefer to show the non-processed reconstruction data so that it is not biased.

5. Throughout: Please make clear that the concentrations reported are in units of at.%, or clarify when that is not the case

Response

We have clarified this at the first mention of a composition: “the Cu segregation was easily isolated using a 1.8 at. % Cu isosurface (bulk Cu content is 0.4 at. %, all subsequent compositions are in atomic percent and will be referred to by % for brevity).”

6. Page 3 and elsewhere: “isosurface” should be replaced by “isoconcentration surface” which is the more precise term

Response

We have done this in the manuscript and the Supporting Information so that the most precise term is used.

7. Section S5: Add representative mass spectra from all four materials/conditions. Log-scale preferable. If the ranging/peak identifications are different please also include that information.

Response

We have added representative mass specs for each sample along with peak identification. Additionally, we show mass specs before and after removal of the Ga rich region in order to show how that influences the analyzed data. We have also explained ranging of the peaks in the Supporting Information. “All peaks were ranged in the same location between samples, but the width varied based on the full width at half maximum (FWHM) of the peak and the thermal tails. The peaks were ranged to approximately two times the background unless another peak was present before this occurred.”

8. Section S5: Isosurface analysis: Clearly define your selected values for voxel size and delocalization so that others can replicate your results.

Response

We generally used the default settings in IVAS (3 x 3 x 3 nm delocalization) and 1 nm voxel size). However, there were instances in which the voxel size was reduced to create a higher polygon density to increase the SNR because the edge polygons are not included in the proximity histogram analysis. The delocalization remained the same, so the shape of the interfaces was not significantly altered with the reduced voxel size. We have added this discussion to the Supplementary Information.

REVIEWERS' COMMENTS:

Reviewer #1 (Remarks to the Author):

I have read the authors' response to my comments and the other two reviewers of the manuscript. The authors have responded to most of my criticisms of the original report, but I am afraid I still do not find their conclusions important to the problem of deactivation of Cu-SSZ-13 zeolites. It is very clear that the APT technique can clearly differentiate the differences in structural evolution between Cu-ZSM-5 and Cu-SSZ-13 zeolites. But this was already known.

The really important problem is what is the mechanism of deactivation of Cu-SSZ-13 and I am not convinced this report adds anything significant to our understanding this process. In fact, the last sentence in the report, prior to the Methods section states:

"The findings of this study further reinforce the fundamental mechanisms behind the stability of the CHA framework under demanding tailpipe reaction conditions."

I do not disagree with this statement, but it shows that the findings are not novel. Our understanding of the origin of the stability of SSZ-13 remains the same. Some indications of CuO nanoparticle formation do appear in their UV/vis and APT results, but the aged catalysts are nearly as active as the initial samples and these observed differences have then little catalytic impact, if any, and cannot be extrapolated to ascertain that under more stringent conditions or longer reaction times they are the mechanism of deactivation. I think this report could be published in a more specialized journal such as ACS Catalysis, or perhaps a materials characterization journal, but in this revised form, it does not meet the high standards for publication of Nature Communications.

Reviewer #3 (Remarks to the Author):

The authors have addressed my technical questions regarding their manuscript. I still believe this manuscript will be of interest to the general scientific community and I leave it to the other reviewers to comment further on its potential impact specifically to the catalysis community. The authors have gone to great lengths to justify the validity of their observations using APT and in doing so have compiled one of the most rigorous supplemental data sections I have reviewed in recent memory. I am happy to report that I have found no serious flaws in their data interpretation and presentation. While the APT data is imperfect, the authors are transparent about its shortcomings, particularly in regards to quantitative accuracy both for composition and spatial resolution.

There are a few minor issues in the supplemental data that can be corrected or commented on as the authors see fit.

- Fig S7/S8: The peak identified as O(2+) is not at the correct position and is most likely 12C(2+). The unidentified peak at 12 Da is similarly 12C(1+). This error should not affect any of the results/discussion. Somewhat interesting that these C peaks only occurs in Cu-ZSM-5. The O(2+) found in the Cu-SSZ-13 samples seems fine (and rather interesting as I am unaware of that peak having been observed previously).
- Caption of Fig S9/10: There is a discrepancy in the sample # (sample 3) and its description (fresh Cu-ZSM-5). I believe sample 3 should be aged Cu-ZSM-5, according to Table S1.
- Table S1: The authors state "A background correction was performed for all compositions" but do not state what that background correction procedure was. Even within the IVAS software package there are multiple ways this could be accomplished, so this needs further clarification. This is particularly important when there are obvious differences in the peak shapes between the two

materials that should be handled differently for accurate (even qualitatively accurate) background correction.

- Considering the differences in the mass spectra between the specimens, it would be nice to include SEM images of each corresponding tip shape so others can learn what “worked” and to what extent. This has implications to the data analysis and interpretation here as the Al₂⁺ peak, particularly its thermal hump, is buried in the Si₂⁺ peak for Cu-ZSM-5 but not Cu-SSZ-13. Note that this is the primary Al peak in the mass spectra. This issue is currently side-stepped by the authors by considering trends rather than absolute quantification, but achieving a more reliable mass spectrum from Cu-ZSM-5 would enable a more quantitative approach.
- Figure S22: N-min = 90. I assume this is a typo as that value makes no sense from the corresponding cluster count plots.

REVIEWERS' COMMENTS:

Reviewer #1 (Remarks to the Author):

I have read the authors' response to my comments and the other two reviewers of the manuscript. The authors have responded to most of my criticisms of the original report, but I am afraid I still do not find their conclusions important to the problem of deactivation of Cu-SSZ-13 zeolites. It is very clear that the APT technique can clearly differentiate the differences in structural evolution between Cu-ZSM-5 and Cu-SSZ-13 zeolites. But this was already known.

The really important problem is what is the mechanism of deactivation of Cu-SSZ-13 and I am not convinced this report adds anything significant to our understanding this process. In fact, the last sentence in the report, prior to the Methods section states:

"The findings of this study further reinforce the fundamental mechanisms behind the stability of the CHA framework under demanding tailpipe reaction conditions."

I do not disagree with this statement, but it shows that the findings are not novel. Our understanding of the origin of the stability of SSZ-13 remains the same. Some indications of CuO nanoparticle formation do appear in their UV/vis and APT results, but the aged catalysts are nearly as active as the initial samples and these observed differences have then little catalytic impact, if any, and cannot be extrapolated to ascertain that under more stringent conditions or longer reaction times they are the mechanism of deactivation. I think this report could be published in a more specialized journal such as ACS Catalysis, or perhaps a materials characterization journal, but in this revised form, it does not meet the high standards for publication of Nature Communications.

While we thank the reviewer for the comments we believe that a recent publication in *Science* (doi: 10.1126/science.aan5630), which appeared after we had submitted revisions, further highlights the importance of this work and we have added several sentences to the manuscript to this end. We have also updated Fig. 1 to reinforce the results.

"A recent experimental and theoretical report has demonstrated that NOX SCR with Cu-SSZ-13 falls outside the conventional boundaries of homogeneous or heterogeneous catalysis as the reaction exhibits a density-dependent interaction of multiple ionically tethered single sites.⁵¹ Of importance to the present findings is that optimization of the Cu spatial distribution or mobility is vital to improving low temperature performance, as Cu ions are found to have a maximum diffusion distance of ~0.9 nm due to electrostatic tethering. While a random Cu distribution is normally assumed, we have shown in fresh Cu-SSZ-13 that this is not the case, as Cu rich regions have been identified (Fig. 3) which may have important implications for forming dynamic multinuclear sites, and may also lead for highly heterogeneous reaction behavior within a single zeolite crystal."

"In the CHA framework the small 8 MR pores are too small for Al migration, preventing Al-Cu clustering to the same extent as is possible in the MFI framework with its larger 10 MR pores, greatly limiting the degradation of this material. However, the CHA pores are large enough for Cu migration, recently demonstrated to be vital for low-temperature reactivity, though as there is no significant migration of Al the Cu remains electrostatically tethered and therefore catalytically active, explaining the retained performance of the material since inactive copper aluminate species are not significantly formed.⁵¹"

Reviewer #3 (Remarks to the Author):

The authors have addressed my technical questions regarding their manuscript. I still believe this manuscript will be of interest to the general scientific community and I leave it to the other reviewers to comment further on its potential impact specifically to the catalysis community. The authors have gone to great lengths to justify the validity of their observations using APT and in doing so have compiled one of the most rigorous supplemental data sections I have reviewed in recent memory. I am happy to report that I have found no serious flaws in their data interpretation and presentation. While the APT data is imperfect, the authors are transparent about its shortcomings, particularly in regards to quantitative accuracy both for composition and spatial resolution.

There are a few minor issues in the supplemental data that can be corrected or commented on as the authors see fit.

- Fig S7/S8: The peak identified as O(2+) is not at the correct position and is most likely 12C(2+). The unidentified peak at 12 Da is similarly 12C(1+). This error should not affect any of the results/discussion. Somewhat interesting that these C peaks only occurs in Cu-ZSM-5. The O(2+) found in the Cu-SSZ-13 samples seems fine (and rather interesting as I am unaware of that peak having been observed previously).

Thank you for bringing this to our attention. We have fixed the error on the figures. The peaks were correctly ranged in the analysis in IVAS and the mistake was just in making the figure so it did not affect the analysis.

- Caption of Fig S9/10: There is a discrepancy in the sample # (sample 3) and its description (fresh Cu-ZSM-5). I believe sample 3 should be aged Cu-ZSM-5, according to Table S1.

Thank you for bringing this to our attention, it should be "aged" and we have corrected the caption.

- Table S1: The authors state "A background correction was performed for all compositions" but do not state what that background correction procedure was. Even within the IVAS software package there are multiple ways this could be accomplished, so this needs further clarification. This is particularly important when there are obvious differences in the peak shapes between the two materials that should be handled differently for accurate (even qualitatively accurate) background correction.

We have added further details to the Supporting Information to clarify how the background correction was performed. "The Local Range-Assisted background estimate embedded in IVAS was used, which estimates the background based on the number of counts before and after the ranged peak."

- Considering the differences in the mass spectra between the specimens, it would be nice to include SEM images of each corresponding tip shape so others can learn what "worked" and to what extent. This has implications to the data analysis and interpretation here as the Al₂⁺ peak, particularly its thermal hump, is buried in the Si₂⁺ peak for Cu-ZSM-5 but not Cu-SSZ-13. Note that this is the primary Al peak in the mass spectra. This issue is currently side-stepped by the authors by considering trends rather than absolute quantification, but achieving a more reliable mass spectrum from Cu-ZSM-5 would enable a more quantitative approach.

We have added before and after SEM images of the tips to the Supplementary Information for the 4 samples that have mass specs included.

- Figure S22: N-min = 90. I assume this is a typo as that value makes no sense from the corresponding cluster count plots.

We have reexamined the data and replotted the NND to be order = 10 and replotted the cluster size distribution with log axes so that the data can be more easily visualized. Overall, we removed the quantitative cluster analysis for AI in this material and want to emphasize there is a non-random AI distribution, but not quantify further as we believe this is the most important message and do not want to make it more complicated than is necessary.